# Non-Dairy Fermented Beverages as Potential Carriers to Ensure Probiotics, Prebiotics, and Bioactive Compounds Arrival to the Gut and Their Health Benefits

**DOI:** 10.3390/nu12061666

**Published:** 2020-06-03

**Authors:** Estefanía Valero-Cases, Débora Cerdá-Bernad, Joaquín-Julián Pastor, María-José Frutos

**Affiliations:** 1Research Group on Quality and Safety, Food Technology Department, Miguel Hernández University, 03312 Orihuela, Spain; e.valero@umh.es (E.V.-C.); dcerda@umh.es (D.C.-B.); 2Engineering Department, Miguel Hernández University; 03312 Orihuela, Spain; jjpastor@umh.es

**Keywords:** intestinal microbiota, vegetable drink, fermentation, beneficial microorganisms, lactic acid bacteria, cereal, legume, pseudocereal, fruit, synbiotic

## Abstract

In alignment with Hippocrates’ aphorisms “Let food be your medicine and medicine be your food” and “All diseases begin in the gut”, recent studies have suggested that healthy diets should include fermented foods to temporally enhance live microorganisms in our gut. As a result, consumers are now demanding this type of food and fermented food has gained popularity. However, certain sectors of population, such as those allergic to milk proteins, lactose intolerant and strict vegetarians, cannot consume dairy products. Therefore, a need has arisen in order to offer consumers an alternative to fermented dairy products by exploring new non-dairy matrices as probiotics carriers. Accordingly, this review aims to explore the benefits of different fermented non-dairy beverages (legume, cereal, pseudocereal, fruit and vegetable), as potential carriers of bioactive compounds (generated during the fermentation process), prebiotics and different probiotic bacteria, providing protection to ensure that their viability is in the range of 10^6^–10^7^ CFU/mL at the consumption time, in order that they reach the intestine in high amounts and improve human health through modulation of the gut microbiome.

## 1. Introduction

Following Hippocrates’ aphorism “Let food be your medicine and medicine be your food”. In recent years, consumers have become more aware of the relationship between health and diet, and are now demanding healthier products with better nutritional characteristics and specific components to prevent health problems and improve their quality of life and life expectancy. This trend offers new opportunities for products with health benefits beyond basic nutrition, which meet consumer expectations and at the same time, drives the growth of the functional food market [1,2,3]. Nowadays, functional foods have been developed that incorporate different components with health benefits, such as bioactive compounds isolated from plants, polyunsaturated fatty acids, probiotics, prebiotics, minerals and vitamins, among others [4,5,6,7].

Twenty-five centuries ago, Hippocrates also stated that “All diseases begin in the gut” when referring to the relevance of the gastrointestinal system for human health. Currently, scientists studying the human microbiome suggest that healthy diets should include fermented foods to temporally enhance live microorganisms in our gut. The intestinal microbiota is a huge and varied collection of microorganisms. The large intestine hosts around 10^13^–10^14^ microorganisms (almost 10 times the number of cells that make up the human body) and most consist of the bacteria phyla Firmicutes and Bacteroidetes. They play an important role in the health of the host, having effects on the regulation of energy metabolism and maturation of the immune system [8]. The administration of live probiotics maintains the balance of gut microbiota and contributes to the overall intestinal health. As a result, fermented foods have gained popularity and are highly demanded by the population [9,10]. Products with probiotics which contribute to gut health represent one of the largest and fastest growing sectors. Probiotics are defined as “live microorganisms that when are administered in an adequate amount confer a beneficial effect on the host health” [11]. Probiotics can be consumed as part of fermented foods or as dietary supplements. The most common genera that have been employed as probiotics and are available on the food market are the lactic acid bacteria (LAB) *Lactobacillus* and *Bifidobacterium*. Their species have mostly been given a generally-recognized-as-safe (GRAS) status and qualified presumption of safety (QPS) status, as their consumption does not present risks for the host’s health [11,12,13].

A probiotic must survive during gastrointestinal digestion and adhere to the intestinal epithelium to exert its beneficial effects. The adherence ability depends on the hydrophobicity and autoaggregation capacity of the probiotic microorganisms [14]. The period of survival and residence of probiotics in the colon can be influenced by their duration and dose of probiotic, as well as by the matrix used as a carrier of the probiotics (Figure 1) [15,16].

Dairy foods have traditionally been used as carriers for probiotic microorganisms. Therefore, foods such as kefir, milk, yogurts, and cheese have been widely explored as dairy matrices for probiotic bacteria [17,18,19,20]. However, certain sectors of the population such as those allergic to milk proteins, those who are lactose intolerant, and those who are strictly vegetarian, cannot consume dairy products. Therefore, a need has arisen to offer consumers an alternative to fermented dairy products by exploring new non-dairy matrices as probiotics carriers [21,22]. However, the viability of probiotic microorganisms is more difficult to maintain in non-dairy matrices than in dairy matrices. The physicochemical parameters must be carefully controlled to guarantee the probiotic viability and to achieve adequate organoleptic properties (mainly aroma and flavor) that can be modified by fermentation [23]. Nevertheless, to improve the probiotic viability in non-dairy beverages, prebiotics could be used as a supplement. Prebiotics can be defined as “non-digestible food ingredients that beneficially affect the host by selectively stimulating the growth and/or activity of one of a limited number of bacteria in the colon” [24]; thus, they can improve the gut microbiome by specific beneficial bacteria fermentation in the colon. Foods or beverages that contain probiotics and prebiotics are known as synbiotic foods. Therefore, the proper selection of food matrices as potential carriers of probiotics is an essential factor to consider in the development of probiotic foods. It is important to ensure the viability of probiotics during processing and storage, in order to maintain their concentrations at high levels (10^6^–10^7^ colony-forming units (CFU) per mL or g of food) at the time of consumption (Figure 2). It is also essential to ensure their survival during gastrointestinal digestion, and thus a high viability of the probiotics, so that a sufficient amount reach the large intestine to exert their beneficial effects. Accordingly, this review aims to explore the potential of different fermented non-dairy beverages (legume, cereal, pseudocereal, fruit, and vegetable), as carriers of bioactive compounds (also generated during the fermentation process), prebiotics and different probiotic bacteria. This review will also examine their effectiveness in providing protection to ensure high probiotic survival during processing, storage and gastrointestinal digestion, in order to make sure that they reach the intestine in sufficient amounts to improve human health.

## 2. Non-Dairy Fermented Beverages as Vehicles for Bioactive Compound, Probiotic, and Prebiotic Delivery to the Gut and their Health Benefits

### 2.1. Fermented Legume Beverages

Several researchers have tried to produce non-dairy probiotic beverages based on legume, as a food matrix for the delivery of bioactive compounds, probiotics, or prebiotics, in order to enhance human intestinal health. Legumes are potential matrices as carriers of probiotics because they contain non-digestible oligosaccharides that can be metabolized by the microorganisms. Soybean legume (*Glycine max*, L.) is the most used since it has high quality proteins and minerals, and due to its isoflavones contents it has the potential to reduce the incidence of osteoporosis and menopausal symptoms [25]. However, due to soy allergies which affect about 0.5% of the general population, other legumes, such as chickpeas (*Cicer arietinum*, L.), are also used [26]. Chickpeas contain a high amount of resistant starch and amylose and some studies have proved that they can reduce the risks of high blood pressure and type-2 diabetes [27].

The production of soy beverages is one of the traditional unfermented food uses of soybeans. Recently, some studies have developed fermented soy drinks with probiotics, in order to improve their beneficial health properties and their flavor and texture [28]. Bedani et al. [29] prepared a soy beverage fermented with probiotic cultures and studied the viability and resistance of probiotics to simulated gastrointestinal digestion, as well as the effect of adding ingredients such as inulin as prebiotic and okara flour, which is a by-product of the soy milk industry. This study concluded that soy milk is a potential food matrix for the delivery of probiotics and prebiotics which could protect them against gastrointestinal juices. The probiotics used, *Bifidobacterium animalis* Bb-12 and *Lactobacillus acidophilus* La-5, survived at high concentrations maintaining their viability at above 8 Log CFU/mL during 28 days of storage at 4 °C with an acidic pH between 4.30 and 4.70. However, the survival was not affected by inulin and/or okara flour. In addition, during the in vitro gastrointestinal digestion, *B. animalis* Bb-12 maintained populations above 7 Log CFU/mL, but *L. acidophilus* La-5 was very sensitive and its viability was reduced, the final concentration being below 5 Log CFU/mL (the initial concentrations of both probiotics were above 8 Log CFU/mL). Therefore, soybeans could be a potential vehicle for probiotics, being able to maintain a high viability to exert their beneficial effects on human gut microbiota, although further studies are required. 

Other studies have combined soy milk with cereals (sprouted wheat, barley, and pearl millet) and legumes (green gram), or with peanut milk, producing beverages with probiotic characteristics and high cell concentrations after fermentation [30,31]. Mridula and Sharma [30] analyzed fermented beverages based on soy milk combined with different cereals or legumes, such as green gram (*Vigna radiata*, L.). In the beverage based on green gram, after the fermentation at 37 °C for 8 h by *L. acidophilus* NCDC14, the probiotic count ranged from 10.36 to 11.32 Log CFU/mL with an acidity between 0.50% and 0.80% and a pH of 4.2–4.4. Besides, a high sensory acceptability score was obtained. Therefore, this fermented legume beverage might be a potential vehicle for probiotics. Santos et al. [31] developed a fermented beverage of soy and peanut milk, using these two substrates to improve the nutrient availability for probiotics. For the fermentation, six different LAB were used, including probiotic strains (*Pediococcus acidilactici* UFLA BFFCX 27.1, *Lactococcus lactis* CCT 0360, *Lactobacillus rhamnosus* LR 32, and *Lactobacillus acidophilus* LACA 4) and yeasts, in a binary culture or in co-culture. *L. acidophilus* LACA 4 and *P. acidilactici* UFLA BFFCX 27.1 reached counts above 8 Log CFU/mL after fermentation for 24 h at 37 °C and another 24 h at ±4 °C.Higher lactic acid contents were also obtained by co-culture with *Saccharomyces cerevisiae* yeast which might serve as a source of vitamin B and proteins. Therefore, a beverage based on peanuts may be a good carrier of probiotics, since it allows larger populations of the probiotic bacteria to be grown and maintained, but further studies are necessary. 

There are few documented studies that have determined the effect of fermented legume beverages on modulation of the gut microbiome in either animal or humaninvestigations. Cabello-Olmo et al. [32] evaluated the impact of long-term supplementation with a non-dairy fermented beverage in the development of type-2 diabetes in rats, which is related to the host’s intestinal microbiota since diabetic individuals present a characteristic intestinal microbial community. The plant-based beverage was composed of legumes and cereals such as alfalfa meal, soya flour, and barley sprouts with other minor components. It was fermented at 37 °C after the incorporation of LAB and debittered brewer’s yeast, and the *Lactobacillus* genus was the most predominant. This research analyzed the fecal microbiota of rat feces collected at six months of study. The specimens had ad libitum access to food supplemented with the fermented beverages whose composition included a high level of fermentable carbohydrates. The study revealed that, at genus level, supplementation with the fermented beverage enriched the abundance of *Sutterella* which includes commensal species found in healthy humans and animals, and *Proteus*, for which there is no evidence for its function in diabetes. However some studies have suggested its role in several pathological conditions. Some *Barnesiella* species and the *Anaerococcus* genus were more numerous in the group treated with the beverage. The abundance of *Barnesiella* is related to a better glucose tolerance and important metabolic improvements, and the relative abundance of *Anaerococcus* includes many bacterial species which produce butyrate in experimental conditions. Butyrate, as well as acetate and propionate, are short-chain fatty acids (SCFAs) which produce beneficial effects on the gastrointestinal tract. SCFAs are produced by the probiotic fermentation of non-digestible carbohydrates and improve the intestinal barrier, inhibiting the development of pathogens and the production of toxic elements, and they are used by intestinal cells as colonocytes to grow [33]. Therefore, the administration of this plant-based beverage fermented with probiotics leads to an enrichment of the gut microbiota population, improving glucose metabolism and protecting against type-2 diabetes development in rats. These novel fermented beverages could be potential vehicles of probiotics, prebiotics, and/or bioactive compounds to protect against metabolic alterations of the diabetic pathology. However, further studies on microbial metabolites which are also important and responsible for gut health are required. 

Another study on the effect of the intake of a soy milk beverage fermented with *Lactobacillus casei* Shirota on gut microbiota in sixty healthy premenopausal women twice a day for 8 weeks, reported that there was an increase of *Lactobacillaceae* and *Bifidobacteriaceae* levels and a decrease of *Enterobacteriaceae* and *Porphyromonadaceae* levels during the intake period. The results suggested that a daily intake of fermented soy milk beverage beneficially contributes to modulation of the gut microbiota in premenopausal healthy women [34]. 

Other studies have focused on the use of different legumes, such as chickpeas (*Cicer arietinum*, L.), as alternatives to soy in fermented plant-based beverages, which could be a promising carrier of probiotics. The results showed probiotic counts of about 6 Log CFU/mL after fermentation for 16 h at 42 °C, but more optimization studies are required to minimize syneresis and improve the sensory acceptability [35]. One study developed a non-dairy beverage fermented by *Lactobacilli acidophilus* probiotics using germinated and ungerminated seeds of moth bean (*Vigna aconitifolia*, L.) and cereals, obtaining promising results since the levels of microorganisms after fermentation for 6 h at 37 °C were above 8 Log CFU/mL and a good sensory acceptability was obtained [36]. The main studies that have been conducted to date on fermented legume beverages and their results are summarized in Table 1. 

### 2.2. Fermented Cereal Beverages

Cereals are consumed all over the world and are considered one of the most important sources of carbohydrates, proteins, dietary fiber, minerals, and vitamins in our diet. Therefore, they are a good option among non-dairy raw materials for producing fermented beverages [37]. Oat (*Avena sativa*, L.) is a potential functional ingredient, due to its proteins, soluble fiber, and antioxidant properties, with β-glucan being the most important carbohydrate fraction because of its prebiotic properties in the gut [38]. Kedia et al. [39] investigated the prebiotic potential of oat through in vitro fermentation for 24 h with human fecal cultures. Their results showed an increase in SCFAs and beneficial intestinal bacteria such as *Enterobacteria*, and a reduction in harmful bacteria such as anaerobes and clostridia. Oat beverages fermented with different *Lactobacillus* strains have also been reported to display a high antioxidant capacity and an increase in polyphenol content with respect to non-fermented beverages. However, only fermented oat with *L. plantarum* LP09 showed an in vitro decrease in the hydrolysis index of starch (used as a measure of the glycemic index in healthy subjects). The fermentation also increased the β-glucan content. However, the soluble and insoluble fiber ratio decreased after fermentation. At the same time, the aroma and flavor were better than in the unfermented control samples [40]. Other studies have also shown that fermented oat beverages may be potential probiotic carriers. They resulted in optimization of the fermentation process and the beverage formulations, reaching high microbial levels (˃10^7^ CFU/mL) during production and storage, and were able to maintain the β-glucan content at the end of storage [41,42,43,44,45]. The viability and in vitro probiotic potential were recently investigated by Funck et al. [46] in oat beverages fermented with *L. curvatus* P99. The results showed a high probiotic viability (above 7 log CFU/mL) during 35 days of being refrigerated at 4 °C. The acceptability of the fermented beverages was good, since most consumers gave high scores for all sensory attributes evaluated. Regarding the probiotic properties, *L. curvatus* P99 showed high gastrointestinal survival and antimicrobial activity against Gram-negative and Gram-positive bacteria. Additionally, it has the capacity of auto-aggregation and of blocking the adhesion of pathogenic bacteria to gut epithelial cells. Johansson et al. [47] carried out an in vivo controlled randomized double-blind study with a rose-hip beverage supplemented with oats fermented with *L. plantarum* DSM9843 in 48 healthy adults. The group receiving the administration of 400 mL/day of this beverage for three weeks showed an increase of total fecal carboxylic acid (lactic, acetic, and propionic acid) and the probiotic bacteria were found in feces at a high concentration, indicating their survival during gastrointestinal digestion. Decreased flatulence and a softer stool consistency were also reported.

Among other cereals, rice (*Oryza sativa*, L.) is also used in the production of fermented beverages that are very popular in Asian-Pacific countries. Ghosh et al. [48] demonstrated a probiotic role of *L. fermentum* KKL1 in a fermented rice beverage. The beverages fermented at 37 °C for 4 days consecutively in anaerobic conditions showed a strong antioxidant activity; the production of glucoamylase, α-amylase and phytase (therefore, an increase of phytate bioavailability) and a mineral increase (Ca, Fe, Mg, Mn and Na). Besides, the hydrosoluble vitamin content in fermented samples was higher than in unfermented control samples such as folic acid, thiamine, ascorbic acid and pyridoxine. However, riboflavin decreased due to bacterial metabolism during fermentation. At the same time, rice was a good carrier for maintaining the *L. fermentum* KKL1 at a high concentration after gastrointestinal digestion and showed sensitivity to the antibiotics tested, except for polymyxin. Nevertheless, this strain showed a moderate cell surface hydrophobicity. Another study reported similar results in fermented rice beverages, but with different species of *Lactobacillus* bacteria, such as *L. plantarum* L7. This strain showed good in vitro characteristics, such as high survival to gastrointestinal digestion, antimicrobial activity, autoaggregation to the intestinal cell surface, and susceptibility to some antibiotics, the latter is a desirable characteristic because probiotics should not carry transmissible antibiotic-resistance genes, in order to avoid the development of new antibiotic-resistant pathogens The fermented beverages increased the lactic, succinic and acetic acids during fermentation for 6 days under anaerobic conditions at 35 °C. At the same time, increases in phytase activity and minerals (Na, Mg, Mn, Fe and Ca) were also observed. Furthermore, the fermented rice beverage presented a high antioxidant capacity [49]. Therefore, rice is a good matrix for *L. fermentum* KKL1 and *L. plantarum* L7 growth and survival in adequate concentrations. Furthermore, these bacteria played an important role in improving the functional properties of rice beverages. However, in vivo investigations are required to explore and verify their probiotic properties. 

Maize (*Zea mays*, L.) is another highly consumed cereal that contains about 72% starch, 10% proteins, and 4% fiber, together with vitamin B and essential minerals [50]. Menezes et al. [51] used the *L. paracasei* LBC-81 with the yeast *S. cerevisiae* CCMA 0731 and *L. paracasei* LBC-81 with *S. cerevisiae* CCMA 0732 in combination, for the fermentation of maize beverages at 30 °C for 24 h. The results showed a high microorganism viability of above 7 Log CFU/mL during fermentation and during 28 days of refrigerated storage (4 °C). The beverages achieved a score of 5 out of 9 points for general acceptance, corresponding to the descriptor “neither dislike nor like”. Lactic acid was the main organic acid produced during the fermentation time and low concentrations of acetic acid and ethanol were also detected. A total of 70 volatile compounds were identified. Although the physicochemical results presented in this study were interesting, more in vivo studies on the viability and health benefits of these microorganisms are needed.

The fermentation process can be used for the delivery of probiotic bacteria and for food detoxification. Probiotic growth in the fermented food medium reduces toxins in raw materials. Aflatoxins can suppress the activity of the human immune system, affect nutrient absorption, and induce liver cancer. This was evaluated by Wacoo et al. [52] in a modification of the traditional method of the production of the Kwete beverage (traditional African fermented maize). Kwete was produced by fermentation with *L. rhamnosus* yoba 2012 and *S. thermophilus* C106 at 30 °C for 24 h. The results showed that the beverage was stable for a month under refrigeration storage at 4 °C, with a mean pH of 3.9 and titratable acidity of 0.6%; the bacteria could also reduce the aflatoxins to undetectable levels during fermentation. The aflatoxins reduction is a novel approach to detoxification of this kind of beverage widely consumed in Africa. 

The use of cereal mixtures for the development of fermented beverages has also been investigated. Rathore et al. [5] made comparisons of single and mixed cereal beverages fermented with different strains of LAB. The results of this study indicated that the organic acid production in mixed cereal substrates was lower than in the single cereal beverages. However, the microbial populations were similar for all substrates. These results are very interesting for future investigations on sensorial properties and consumer acceptance. Nevertheless, malt was the best substrate for microbial growth used as single or mixed beverages. Freire et al. [53] developed mixed beverages from rice and maize fermented with *L. acidophilus* LACA 4 and *L. pantarum* CCMA 0743 for 36 h at 37 °C and supplemented with frutooligosaccharides (FOS). The FOS were an important prebiotic for maintaining the microbial viability in high concentrations (>10^7^ CFU/mL) during 28 days of refrigerated storage (4 °C). The main organic acids were lactic and acetic acid. The sensory acceptance of the beverages was good with high scores with respect to unfermented ones, indicating their high potential for the market.

Maize beverages have been shown to be a potential matrix for the growth and survival of the bacteria studied, with important changes in their composition. However, in the future, further studies focused on the identification, quantification and potential effect of bioactive compounds, as well as the modulation of intestinal microbiota, will be of great interest to elucidate the functionality of maize beverages in the human organism. A summary of the studies on fermented cereal beverages conducted to date and their results is presented in Table 2.

### 2.3. Fermented Pseudocereal Beverages

Several studies have focused on the development of non-dairy probiotic beverages using pseudocereals as vehicles for the delivery of bioactive compounds, probiotics, and prebiotics. Pseudocereals are viable potential substrates, as they contain nutrients easily metabolized by probiotic microorganisms. They are a good source of high-quality proteins comparable to those of cereals, minerals (Ca, Cu, Fe, Mg, Mn and Zn) in higher amounts than in conventional cereals, carbohydrates, and fiber [54]. Quinoa (*Chenopodium quinoa*, L.) is the pseudocereal most widely used as a food matrix. It is the only plant-based food that has all of the essential amino acids (lysine, methionine, and threonine), trace elements and vitamins, and its protein quality to matches that of milk [55]. It can also decrease the risk of type-2 diabetes and cardiovascular diseases [56]. Furthermore, it is gluten-free, so its consumption is suitable for celiacs and people with gluten-allergy problems.

One study reported the use of two varieties of quinoa (Rosada de Huancayo and Pasankalla) as suitable food matrices for the development of fermented beverages [57]. The fermentation was carried out by the probiotics *Lactobacillus plantarum* Q823, *Lactobacillus casei* Q11, and *Lactococcus lactis* ARH74 for 6 h at 30 °C. After 28 days of storage (5–7 °C), *L. plantarum* Q823 and *L. casei* Q11 were detected at levels higher than 9 Log CFU/mL, which is a concentration above the recommended minimum for probiotic effects, with initial concentrations of 8 Log CFU/mL. Vera-Pingitore et al. [58] have previously reported that *L. plantarum* Q823 can survive during the passage through the human gastrointestinal tract. In this study, seven healthy female volunteers consumed 20 mL of quinoa-based beverage containing 9.19 Log CFU/mL on a daily basis or 7 days, and microbial counts were analyzed in feces. Levels between 5 and 7 Log CFU/mL were detected for at least 7 days after the end of the intake. Therefore, quinoa-based fermented beverages contain high amounts of protein, fiber, vitamins and minerals, with probiotics which could exert health benefits on the human gastrointestinal microbiota. However, more long-term human clinical trial studies are needed to demonstrate that these beverages have probiotic properties.

There are few documented studies that have attempted to determine the in vivo effect of fermented pseudocereal beverages on the modulation of the gut microbiome in either animal or human investigations. However, one study has evaluated the impact of several beverages based on aqueous extracts of soy and quinoa with prebiotics (FOS) and/or with probiotics (*Lactobacillus casei* Lc-01) on the human intestinal microbiota, using the Simulator of the Human Intestinal Microbial Ecosystem (SHIME^®^) [59]. The SHIME is a five-stage sequential reactor system simulating the different parts of the gastrointestinal tract in vitro, representing the human gut microbiota [60]. The study of Bianchi et al. [59] reported that a synbiotic beverage fortified with both a probiotic and a prebiotic showed the best beneficial effect on the gut microbiota, as the oligosaccharides used were hydrocolloids, which protected the microorganism. The concentration used in the SHIME was a proportion equivalent to 8 Log CFU/mL in the beverage. *L. casei* Lc-01 of the synbiotic beverage survived the stomach and intestinal conditions and reached the colon, maintaining its functionality, and the concentration of SCFAs was stable during the in vitro gastrointestinal digestion. Furthermore, the growth of several species of *Lactobacillus* spp. and *Bifidobacterium* spp. in the colon were stimulated by the synbiotic beverage. At the same time, the growth of potential enteropathogenic bacteria such as *Clostridium* spp., enterobacteria and other pathogenic bacteria such as *Bacteroides* spp. and *Enterococcus* spp. was reduced. Another positive effect of the beverage on the gastrointestinal tract was the significant decrease in the ammonia ion production. Ammonia can stimulate the development of colon carcinogenesis, since it can affect intestinal cells, changing their morphology and intermediary metabolism by increasing DNA synthesis [61]. Therefore, this beverage based on quinoa and soybean with FOS and fermented by *L. casei* Lc-01 positively modulates the gut microbiota improving the diversity and richness of beneficial bacteria without affecting their functionality and reducing the growth of the pathogenic ones, and decreasing the production of toxic elements such as ammonia. 

Some studies have investigated the use of other pseudocereals, such as chia (*Salvia hispanica*, L.), amaranth (*Amaranthus*, L.) or buckwheat (*Fagopyrum esculentum*, L.), as food carriers to develop pseudocereal probiotic beverages, obtaining interesting results. In a study on the elaboration of a beverage with the probiotic *Lactobacillus rhamnosus* GG (5–6 Log CFU/mL), using mashed buckwheat previously fermented with LAB, the results showed that the levels of the microorganism were higher than 6 Log CFU/mL after 14 days of cold storage at 6 °C [62]. Kocková and Valík [63] also produced beverages based on buckwheat or dark buckwheat fermented with *L. rhamnosus* GG ATCC 53,103 (6 Log CFU/mL) at 37 °C for 10 h. This probiotic was able to grow and metabolize buckwheat and dark buckwheat and to survive during 21 days of a refrigerated storage period at 5 °C, with the probiotic counts being above the minimum recommended (>7 Log CFU/mL). Kocková et al. [64] studied the use of different pseudocereals as substrates for the production of different beverages fermented with *L. rhamnosus* GG (5 Log CFU/mL) for 10 h at 37 °C, using amaranth flour, amaranth grain, buckwheat flour, or whole buckwheat flour. The results showed that the probiotic could grow and metabolize these pseudocereals and after 21 days of storage at 5 °C, the probiotic levels were higher than 6 Log CFU/mL, and thus over the limit required for probiotic food, except for the beverage with whole buckwheat flour. Another study reported that chia can be fermented by *L. plantarum* C8, and that after 24 h of fermentation, its overall characteristics were improved, such as the phenolic compound concentration and antioxidant activity [65]. Therefore, this pseudocereal could be a good matrix for the development of probiotic beverages. The main studies conducted to date on fermented pseudocereal beverages and their results are summarized in Table 3.

### 2.4. Fermented Fruit and Vegetable Beverages

Fruit and vegetable beverages are an excellent source of vitamins, antioxidants, minerals, and bioactives. At the same time they represent a good alternative to dairy matrices and a good choice for the entire human population, because they have hydration properties, are refreshing and have attractive flavors [21,66]. Therefore, different fruits and vegetable juices are used to develop fermented beverages in combination or alone as an alternative to fermented dairy products. The fermentation process can increase the shelf life of fruit and vegetable beverages, improving their nutritional and functional properties, with beneficial effects on health [16]. Recently, a wide variety of research has been focused on the production of fermented non-dairy synbiotic beverages, including different types of vegetables or fruits, such as blended carrot-orange juices and nectars with different inulin concentrations [67,68], pomegranate juices, and Cornelian cherry beverages using delignified wheat bran [69,70], clarified apple juice with oligofructose [71], orange juice with oligofructose [72], orange juices and hibiscus tea mixed beverage with oligofructose [73], and blended red fruit beverages (strawberry, blackberry and papaya) supplemented with three separate prebiotics: FOS, inulin and galactooligosaccharides [74]. Generally, the findings have indicated a good compatibility among prebiotic ingredients and vegetable beverage matrices. Furthermore, prebiotic supplementation can improve the viability of the different probiotic strains so that they are above the minimum concentration recommended during the beverages processing and storage and are also able to survive during gastrointestinal digestion in order to reach the colon, promoting the growth of beneficial bacteria. Furthermore, a recent randomized, controlled, triple-blinded, parallel trial study of polycystic ovarian syndrome (POCS) showed the effect of synbiotic pomegranate juice (containing inulin and three species of *Lactobacillus*) in terms of improving the testosterone level, body mass index, insulin, insulin resistance, weight, and waist circumference in POCS. However, neither group showed a significant change in the fasting blood sugar, luteinizing hormone, and follicle-stimulating hormone [75].

Several LAB can biotransform polyphenols into phenolic compounds with an improved bioavailability and bioactivity during the fermentation time. Several studies have investigated phenolic compound biotransformation during fermentation and gastrointestinal digestion. The studies on fermented fruit or vegetable beverages have shown an improved antioxidant capacity and phenolic composition modification. Apple juice fermented with *L. plantarum* ATCC14917 at 37 °C improved the antioxidant capacity by increasing the quercetin, phloretin and 5-O-caffeoylquinic acid contents during 24 h of fermentation [76]. Yang et al. [77] reported an increase of the total flavonoids content as well as the antioxidant activity in fermented mixed beverages from apples, carrots, and pears during fermentation with two commercial *L. plantarum* 115 *L. plantarum* Vege Start 60. On the other hand, the biotransformation of phenolic compounds during fermentation and gastrointestinal digestion in fermented pomegranate juices ensured the survival of *L. plantarum* CECT220, *L. acidophilus* CECT903, *B. longum subsp. infantis* CECT4551, and *B. bifidum* CECT870, suggesting a prebiotic effect [78]. Furthermore, one study on a fermented tea infusion found an increase in the overall antioxidant capacity, a modification in the phenolic composition and an increase in their cellular uptake after in vitro digestion [79]. Oolong tea polyphenols have been reported to improve host health through the generation of SCFAs and modulation of the human gut microbiota, leading to potential applications for anti-obesity therapies [80]. Specifically, (-)-Epigallocatechin 3-O-(3-O-methyl) gallate showed a prebiotic effect with the modulation of gut microbiota and obesity prevention in high-fat diet-fed mice [81,82]. Phenolic compounds are recognized as antioxidants, but some of them also exhibit antimicrobial activity. Cueva et al. [83] showed that phenolic compounds generated from probiotic metabolism (phenylpropionic, benzoic and phenylacetic) can inhibit the growth of intestinal pathogens and prevent intestinal dysbiosis.

The ability of probiotic microorganisms to metabolize phenolic compounds is known to depend on the species or strains. However, differences in the total polyphenol content and antioxidant capacity have been shown between different vegetable beverages for the same probiotic strains. These differences may be related to the variability in the phytochemical composition of the different vegetable and fruit matrices [84,85]. In addition, the matrix also has an influence on the exopolysaccharides (EPS) production during fermentation, improving the consistency and antioxidant capacity of fermented juices [86]. At the same time, the EPS also have a significant role as prebiotics and can enhance probiotic colonization in the gut. They have also been used as immunomodulatory, immunostimulatory, antidiabetic, and hypocholesterolemic agents [87,88]. 

Bearing in mind the nutritional importance of fruit and vegetable beverages, some of them, such as fresh prickly pear juices, present hazardous volatile components in negligible quantities. Therefore, the reduction of risky compounds by modification or decomposition during the fermentation time is an interesting strategy. Panda et al. [89] demonstrated prickly pear quality enhancement by fermentation with *Lactobacillus fermentum* ATCC 9338 for 48 h at 28 °C. The study demonstrates the decomposition of several risky organic compounds present in the fresh juice, such as 2-propenenitrile, 2-(acetyloxy); furfuryl alcohol; acetaldehyde; 2,2-diethyl-3-methyloxazolidine; 4h-Pyran-4-one; 3,5-dihydroxy-2-methyl; and furan.

Beyond the nutritional and physicochemical advantages, recent studies have shown that different fermented fruit and vegetable beverages also have some physiological functions. For example, some studies have shown a stable α-glucosidase inhibitory activity with an anti-hyperglycemic in vitro effect in a pumpkin beverage fermented by *L. mali* K8 [90]. Gamboa-Gómez et al. [91] also showed an anti-hyperglycemic effect with an infusion of oak leaves and fermented beverages from *Quercus convallata* and *Q. arizonica* in vitro and in vivo studies with female mice. On the other hand, blended fermented blueberry pomace by *L. rhamnosus* GG, *L. plantarum*-1, and *L. plantarum*-2 showed in in vitro hypocholesterolaemic effect. In addition, this fermented beverage exhibited an outstanding performance in terms of anti-fatigue in a mouse weight swimming experiment [92]. According to Harima-Mizusawa et al. [93], citrus beverage fermented with *L. plantarum* YIT0132 had a good effect in relieving the perennial allergic rhinitis symptoms in a double-blind, placebo-controlled trial. Other fermented beverages, such as those of tomato, feijoa, blueberry-blackberry, cactus pear, and prickly pear fruits exhibit a great in vitro anti-inflammatory capacity and help maintain the integrity of intestinal barrier [85,86,94]. However, the results are influenced by the different vegetable beverage matrices. For example, Valero-Cases et al. [94] showed the best improvement of the intestinal barrier with fermented tomato juices with respect to fermented feijoa ones. The vasorelaxant capacity was proposed for fermented jabuticaba berry beverages through an in vivo study of vascular reactivity in male Wistar rats. Theses beverages could act as an interesting cardiovascular protector [95]. Cheng et al. [96] reported an increase of SCFAs production and an improvement of the fecal microbiota community structure in vitro with blueberry pomace fermented by *L. casei* CICC20280. Wang et al. [97] showed that the consumption of fermented beverages of Changbai Mountain vegetables and fruits can reduce the Firmicutes/Bacteroidetes ratio and increase the Bacteroidales S24–7 group, Bacteroidaceae, the genus *Bacteroides*, and Prevotellaceae in a mouse model study. However, these are results from in vitro and in vivo studies with animal models, so future research in humans is required to evaluate the physiological effects on health improvement. A summary of studies conducted to date on fruit and vegetable fermented beverages and their results is shown in Table 4.

## 3. Conclusions and Future Perspectives

The studies carried out to date have provided useful information and a deeper understanding of the metabolic mechanisms of growth, probiotic viability, and the microbial biotransformation or production of bioactive compounds in fermented non-dairy beverages. Non-dairy matrices (legumes, cereals, pseudocereals, fruits, and vegetables) represent potential carriers of probiotics, prebiotics, and bioactive compounds. They are a good alternative to dairy matrices because it has been proven that the fermentation of these vegetable matrices can improve the shelf life and their safety due to the organic acids generated during the fermentation period, their nutritional and functional composition, and their digestibility. Moreover, in all the matrices reviewed, the probiotic concentrations are above the minimum recommended (>7 Log CFU/mL). Therefore, they are a good alternative to the dairy products on the market that can also be consumed by people intolerant or allergic to milk proteins, those who are hypercholesterolemic, or those who are vegetarian, among others. However, to corroborate the health benefits of fermented non-dairy beverage consumption, further in vivo research, including human clinical studies addressing matrix combinations and doses in different populations, is needed.

## Figures and Tables

**Figure 1 nutrients-12-01666-f001:**
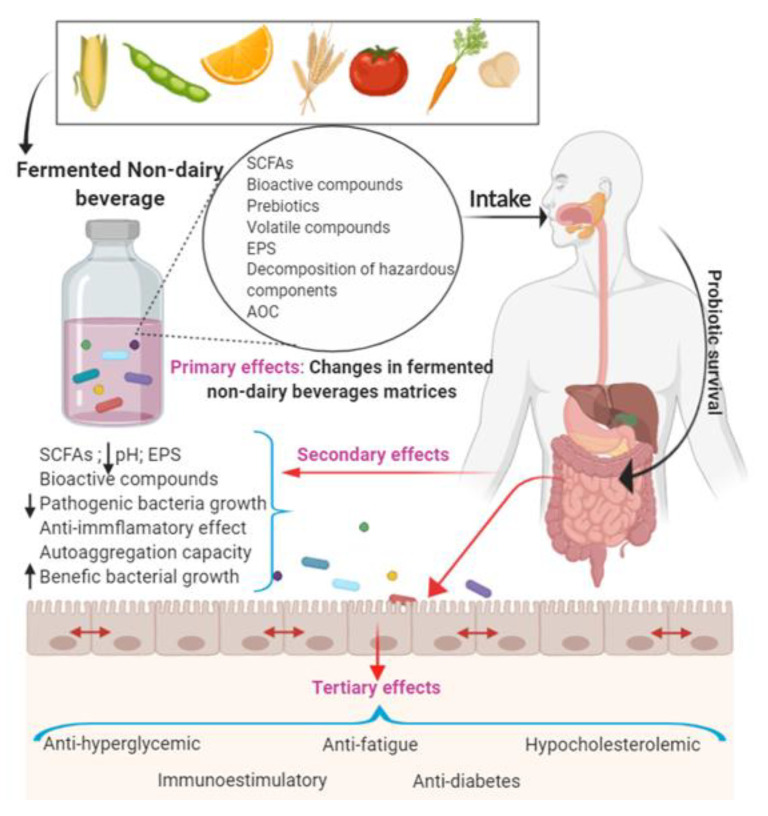
Summary of the beneficial effects of probiotics in different fermented non-dairy beverage matrices. Primary effects: changes in non-dairy matrices during fermentation. Secondary effects: changes in the intestinal epithelium. Tertiary effects: positive changes in health. SCFAs: short chain fatty acids; EPS: exopolysaccharides; AOC: antioxidant capacity [14,15,16].

**Figure 2 nutrients-12-01666-f002:**
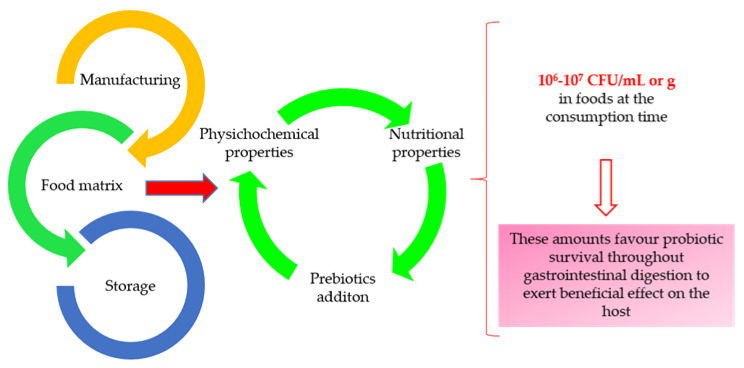
Important factors to be considered when assessing afermented non-dairy beverage matrices as potential carriers for the viability of probiotic bacteria.

**Table 1 nutrients-12-01666-t001:** Summary of fermented legume beverages as potential carriers for bioactive compound, probiotic, and prebiotic delivery to the gut.

Legume Fermented Beverage	Probiotic Bacteria	Results	Reference
Soy milk with inulin and okara flour	*L. acidophilus* La-5, *B. animalis* Bb-12	Probiotic viability above the minimum recommended after 28 days of storage, high probiotic viability after in vitro gastrointestinal digestion	Bedani et al. [29]
Soy milk with green gram	*L. acidophilus* NCDC14	High probiotic viability after fermentation, high sensory acceptability score	Mridula and Sharma [30]
Peanut-soy milk	*P. acidilactici* UFLA BFFCX 27.1, *L. lactis* CCT 0360, *L. rhamnosus* LR 32, L. acidophilus LACA 4	High probiotic viability after fermentation, high acid lactic contents	Santos et al. [31]
Soya flour, alfalfa meal, barley sprouts	Different LAB, mostly Lactobacillus genus	Enrichment of the gut microbiota population	Cabello-Olmo et al. [32]
Soy milk	*L. casei* Shirota	Beneficial modulation of the gut microbiota	Nagino et al. [34]
Chickpeas	*S. thermophilus*, *L. bulgaricus*, *L. acidophilus*	High probiotic viability after fermentation	Wang et al. [35]
Germinated and ungerminated cereals and legumes: barley, ragi, moth bean, soybean	*L. acidophilus*	High probiotic viability after fermentation, good sensory acceptability	Chavan et al. [36]

**Table 2 nutrients-12-01666-t002:** Summary of fermented cereal beverages as potential carriers form bioactive compounds, probiotics, prebiotics delivery to de gut.

Cereal Fermented Beverage	Probiotic Bacteria	Results	Reference
Oat	Human fecal cultures	Increase in SCFAs ^1^, increase of healthy intestinal bacteria, reduction of harmful bacteria	Kedia et al. [39]
Oat	Different *Lactobacillus* strains	High levels of antioxidant capacity, high polyphenols content, increase of β-glucan content during fermentation, decrease in the hydrolysis index of starch, high probiotic viability	Luana et al. [40], Bernat et al. [41], Gupta et al. [42], Gupta and Bajaj [43], Kedia et al. [44], Wang et al. [45]
Oat	*L. curvatus* P99	High probiotic viability after in vitro digestion, antimicrobial activity, blocking of the adhesion of pathogenic bacteria to epithelial cells, autoaggregation capacity	Funck et al. [46]
Oat	*L. plantarum* DSM9843	In vivo study: increase of lactic, acetic and propionic acid, high probiotic viability after digestion; decrease in flatulence; and softer stool consistency	Johansson et al. [47]
Rice	*L. fermentum* KKL1	Strong antioxidant capacity, glucoamylase and α-amylase production, phytase activity, high hydrosoluble vitamins, antibiotic susceptibility	Ghosh et al. [48]
Rice	*L. plantarum* L7	High antioxidant capacity; increase in lactic, succinic, and acetic acid during fermentation; high probiotic viability after in vitro digestion; antibiotic susceptibility; antimicrobial activities; increase in minerals and phytase activity	Giri et al. [49]
Maize	*L. paracasei* LBC-81 with *S. cerevisiae* CCMA 0731 and L. *paracasei* LBC-81 *S. cerevisiae* CCMA 0732	Probiotic viability above the minimum recommended, high production of lactic acid during fermentation, 70 volatile compounds identified	Menezes et al. [51]
Kwete	*L. rhamnosus* yoba 2012 and *S. thermophilus* C106	Decrease in aflatoxins content = detoxification of beverage	Wacoo et al. [52]
Cereals mixing	Mixed or single LAB	Malt was the best substrate for microbial growth used as single or mixed beverages	Rathore et al. [5]
Rice and maize	*L. acidophilus* LACA 4 and *L. pantarum* CCMA 0743 supplemented with FOS	High probiotic viability, increase in lactic and acetic acids during fermentation, good sensorial acceptance	Freire et al. [53]

^1^ SCFAs: short chain fatty acids.

**Table 3 nutrients-12-01666-t003:** Summary of fermented pseudocereal beverages as potential carriers for bioactive compound, probiotic, and prebiotic delivery to the gut.

Pseudocereal Fermented Beverage	Probiotic Bacteria	Results	Reference
Two quinoa varieties (Rosada de Huancayo, Pasankalla)	*L. plantarum* Q823, *L. casei* Q11, *L. lactis* ARH74	Probiotic viability above the minimum recommended after 28 days of storage	Ludena-Urquizo et al. [57]
Quinoa	*L. plantarum* Q823	High probiotic viability after the gastrointestinal digestion	Vera-Pingitore et al. [58]
Aqueous extracts of soybean and quinoa grains with FOS	*L. casei* Lc-01	Positive modulation of the gut microbiota, decrease in toxic elements (ammonia)	Bianchi et al. [59]
Mashed buckwheat previously fermented with LAB	*L. rhamnosus* GG	High probiotic viability after 14 days of cold storage, good sensorial acceptance	Matejčeková et al. [62]
Buckwheat, dark buckwheat	*L. rhamnosus* GG	Probiotic viability above the minimum recommended after 21 days of cold storage	Kocková and Valík [63]
Amaranth flour, amaranth grain, buckwheat flour, whole buckwheat flour	*L. rhamnosus* GG	High probiotic viability after 21 days of storage, except for the beverage with whole buckwheat flour	Kocková et al. [64]

**Table 4 nutrients-12-01666-t004:** Summary of fruit and vegetable fermented beverages as potential carriers form bioactive compounds, probiotics, prebiotics delivery to de gut.

Fruits and Vegetables Fermented Beverage	Probiotic Bacteria	Results	Reference
Synbiotic beverages: Carrot–orange juices and nectars + inulin	Different LAB ^1^ strains	Prebiotic ingredients + vegetable beverage matrices = good compatibility Prebiotic supplementation = viability of probiotic strains above the minimum recommended. High viability of probiotic strains after gastrointestinal in vitro digestion	Valero-Cases and Frutos [67,68], Mantzourani et al. [69], Mantzourani et al. [70], Pimentel et al. [71], Miranda et al. [73], Bernal-Castro et al. [74]
Orange + hibiscus tea + oligofructose			
Red fruit beverage + FOS, GOS, Inulin			
Pomegranate + Cornelian cherry + delignified wheat bran			
Clarified apple juice + oligofructose			
Pomegranate + inulin	Three species of *Lactobacillus*	In vivo study: improve testosterone level, insulin, insulin resistance, body mass index, weight and waist circumference in polycystic ovarian syndrome	Esmaeilinezhad et al. [75]
Apple juice	*L. plantarum* ATCC14917	Improved antioxidant capacity, increasing quercetin phloretin and 5-O-caffeoylquinic acid contents	
Mixed beverages from apples, carrots and pears	*L*. *plantarum* 115 and Vege Start 60	Increase of total flavonoids content and the antioxidant activity	Yang et al. [77]
Pomegranate juices	*L.plantarum* CECT220, *L.acidophilus* CECT903, *B.longum subsp. infantis* CECT4551, *B.bifidum* CECT870	The biotransformation of phenolic compounds during fermentation and gastrointestinal digestion, suggesting a prebiotic effect	Li et al. [76]
Tea infusion	*L. plantarum* ASCC276, *L. plantarum* ASCC292, *L. acidhophilus* CSCC2400, *L.plantarum* WCFSI, *L. rhamnosus* WQS, *L. brevis* NPS-QW145	Increase in the antioxidant capacity, modification of the phenolic composition, increase in cellular uptake after in vitro gastrointestinal digestion	Zhao and Shah [79]
Oolong tea	*Fecal bacteria*	Improved the host health generating SCFAs ^2^ and modulating the human gut microbiota, anti-obesity therapy	Zhang et al. [80]
Prickly pear	*Leuconostoc mesenter**oides* strains	Production of EPS ^3^, improvement of consistency and antioxidant capacity	Di Cagno et al. [86]
Prickly pear	*L. fermentum* ATCC 9338	Decomposition of some risky organic compounds present in the fresh juice like: 2-propenenitrile, 2-(acetyloxy); furfuryl alcohol; acetaldehyde; 2,2-diethyl-3-methyloxazolidine, 4h-Pyran-4-one, 3,5-dihydroxy-2-methyl and furan	Panda et al. [89]
Pumpkin	*L. mali* K8	α-glucosidase inhibitory activity with anti-hyperglycemic effect	Koh et al. [90]
Infusion of oak leaves	Kombucha culture	Anti-hyperglycemic effect and antioxidant activity	Gamboa-Gómez et al. [91]
Blueberry pomace	*L. rhamnosus* GG. *L. plantarum*-1 and *L. plantarum*-2	Hypocholesterolemic and anti-fatigue effect	Yan et al. [92]
Citrus	*L. plantarum* YIT0132	Good effect in relieving perennial allergic rhinitis symptoms	Harima-Mizusawa et al. [93]
Tomato, feijoa, blueberry-blackberry, cactus pear, and prickly pear fruits	Different LAB strains	Great in vitro anti-inflammatory capacity and help to maintain the integrity of intestinal barrier	Di Cagno et al. [86], Valero-Cases et al. [94], Filannino et al. [85]
Blueberry pomace	*L. casei* CICC20280	Increase of SCFA production and an improvement of fecal microbiota	Cheng et al. [96]
Changbai Mountain vegetables and fruits	Naturalized species of bacteria	Reduction of Firmicutes/Bacteroidetes ratio, increase of Bacteroidales S24–7 group, Bacteroidaceae, genus *Bacteroides* and Prevotellaceae in a mouse model study	Wang et al. [97]

^1^ LAB: lactic acid bacteria; ^2^ SCFAs: short chain fatty acids; ^3^ EPS: exopolysaccharides.

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
