# Peer review of "Non-Dairy Fermented Beverages as Potential Carriers to Ensure Probiotics, Prebiotics, and Bioactive Compounds Arrival to the Gut and Their Health Benefits"

_nutrients, 2020, doi:10.3390/nu12061666_

Round 1

Reviewer 1 Report

T The review "Non-dairy fermented beverages as potential carriers to ensure probiotics, prebiotics and bioactive compounds arrival to the gut and their health benefits" it gives informations on interesting and alternative matrices for probiotic microrganisms and derived products. However taking into account the importance of a review to have structured informations I would like to suggest to the authors to organize with an order all the infomations provided. In particular, into every paragraph relating the different matrices (cereals, legumes, etc) they can organize the argument following a ierarchical order regarding involved probiotic microrganism (bacteria, fungi, yeast), prebiotics added, positive effectc on gut and perspectives.
Perspectives could be anticipate from analysy SWOT regarding food security and food safety, by a dedicated table.
Then, the authors have to change the enfasis of conclusions and perspectives, otherwise in the present form the risk is to mine the importance of this topic. In fact, a lot of things have to be analyze and studied before to can promote this kind of beverages. Finally, the latin name of micorganisms needs to to be cecked carefully, and I do not undertand what it means non-bitter beer yeast.
At the light of these considerations I suggest to change the title in "Vegetable fermented beverages as potential carriers of probiotics, prebiotics and bioactive compounds to the gut and their health benefits".

Reviewer 2 Report

The manuscript presents a suitable and target review of the subject, however additional editing to improve the manuscripts grammar is required.

Reviewer 3 Report

The MS on the whole is informative, logically laid out and easy to read.    It does contain a number of minor grammatical errors which need to be corrected.

The authors had a tendency to on occasion use sweeping statements such as on L 98 to 100.  Soybean …… can prevent some diseases such as cancer, osteoporosis and menopausal symptoms ….. .   I do not believe  that there is sufficient evidence to support such a claim and some modifying words such …. “potential to reduce the incidence” should be included here and elsewhere.

In numerous places data on the survival of probiotic bacteria is presented but it  is often incomplete – either starting numbers are provided or final numbers.  What is required is information on – product (this is provided), formulation (as much information as available  – eg pH?), storage conditions (temperature, gaseous atmosphere – VP?) and  storage time, only then can the reader put  data on the final number of cells surviving in the product into context.  Ideally Dvalues or survival data should also be included if provided by the original authors. 

Eg L 199 – simply stating … a high probiotic viability during 35 days (good that the length of the trial is mentioned) of refrigerated (also good that temperature is mentioned) storage  ….. is not that useful .. What does good signify ??

I found the Tables difficult to understand.  What does an up arrow mean in the context of probiotic viability ?  is it an increase in their viability ????? compared to what – presumably the arrow indicates ? an acceptable viability ???.  Table 1 – it is hard to imagine that viability can be increased after 28 days ??.  Same with sensorial acceptance – increased over time?

Table 3.  I can understand the down arrow as indicating that the probiotic decreased flatulence.  However what is meant by a down arrow beside antibiotic resistance – is this a good or a bad thing ??  I think that the data the authors wish to portray cannot simply be simplified into up or down arrows.

The authors also had the tendency to add in single sentence phrases or concepts without explaining their relevance e.g. L 219 ..Nevertheless, this strain showed a moderate cell surface hydrophobicity.   Soooo what ?   was is the point of this statement?  As far as I can tell it is not linked to the proceeding or preceding sections. 

Also L 221 to 222 – the authors need to explain why all of the invitro characteristics are “good”  for example – why is antibiotic susceptibility good in a probiotic ?  why is auto-aggregation to the cell surface good.  If this review is aimed at a general food science or microbiological audience then care needs to taken with regard to assumed knowledge.

On the positive side the section in  L 309 to 312 – is a  good example  - in that a decrease in ammonia ions was reported, followed by an explanation of why this is important.

Unless I missed it, there appeared to be no mention of oral probiotics.   Has no research been carried out on non-dairy oral probiotics ?  if not is this in itself not worth mentioning.  Formulating Oral probiotics have many similar challenges to gut probiotics with regard to survival, but also differences – such as short residence time of most foods in the oral cavity.  Is this an opportunity for non-dairy probiotics?

The Conclusion section seemed quite weak.   All through the main text the authors state more research is needed and this also seemed to be the main thrust of the conclusions.   Why are non-dairy matrices good alternatives ???  What needs to be done to move this field of research forward.

Minor:

L 142.  What is meant by non-bitter beer yeast ?  also as written the sentence dos not make sense ?  LAB are not yeast.
